# Molybdenum Oxide and Nickel Nitrate as Cooperative Sintering Aids for Yttria-Stabilized Zirconia

**DOI:** 10.3390/ma13122875

**Published:** 2020-06-26

**Authors:** Clay Hunt, John Kyle Allemeier, David Driscoll, Adam Weisenstein, Stephen Sofie

**Affiliations:** 1Mechanical & Industrial Engineering Department, Montana State University, Bozeman, MT 59717, USA; kyle.allemeier@gmail.com (J.K.A.); ddriscoll@glacigen.com (D.D.); ssofie@montana.edu (S.S.); 2ZAF Energy Systems Inc., Bozeman, MT 59718, USA; adam.weisenstein@zafsys.com

**Keywords:** sintering, yttria-stabilized zirconia, nickel, molybdenum, dilatometry, X-ray diffraction

## Abstract

The entirely accidental observation of increased sintering performance of nickel-infiltrated yttria-stabilized zirconia (8YSZ) in a molybdenum and oxygen rich atmosphere was explored. Molybdenum and nickel were found to be synergistic sintering aids for 8YSZ. However, sintering had to take place in an atmosphere of flowing oxygen. Samples sintered in air consistently burst. The sintering performance, microstructure, and crystal structure of 8YSZ with additions of both Mo and Ni together are compared to the sintering performance, microstructure, and crystal structure of pure 8YSZ, 8YSZ with only Ni added as a sintering aid, and 8YSZ with only Mo added as a sintering aid. Enhanced densification and grain growth is observed in the Mo–Ni 8YSZ samples when compared to all other sintering samples. Order of magnitude sintering rate increases are observed in the Mo–Ni 8YSZ over that of pure 8YSZ. With a maximum sintering temperature of 1200 °C and a one-hour dwell, sintered densities of 85% theoretical density (5.02 g⁄cm^3^) are achieved with the Mo–Ni samples: a 57% increase in density over pure 8YSZ sintered with the same sintering profile. EIS results suggest conductivity may not be negatively impacted by the use of these two sintering aids at temperatures above 750 °C. Finally, the spontaneous generation of nickel-molybdenum nano-rods was observed on the 5, and 10 mol.% Mo–Ni infiltrated 8YSZ samples after being left under vacuum in a scanning electron microscope chamber, suggesting evaporation of a possible nickel–molybdenum compound from the sample fracture surfaces.

## 1. Introduction

The structural stabilization of the cubic and tetragonal polymorphs of zirconium oxide with the addition of yttrium has yielded an incredibly versatile ceramic with numerous functional and structural applications. This material is currently used to produce kitchen knives, mechanical bearings, artificial gemstones, and medical and dental implants. More technically, yttria-stabilized zirconia (YSZ) is used in the production of thermal barrier coatings of turbine blades, oxygen sensors in fuel-injected engines, and electrolytes and anodes of solid oxide fuel cells (SOFCs) [1,2,3]. The tetragonal phase of zirconia is mechanically tough and is the phase of the polymorphic material used to make bearings, knives, and dental implants [4]. Single crystals of the cubic phase of YSZ are quite transparent, and are used to make artificial diamonds. The low electronic conductivity and the relatively high oxygen ion conductivity make polycrystalline cubic zirconia an excellent candidate for the SOFC electrolyte [5,6].

Zirconia chemically stabilized with 8 mol.% yttria (8YSZ) exhibits a maximum in oxygen ion conductivity with respect to yttria content [7]. 8YSZ also exhibits chemical stability in the caustic fuel cell environment at typical fuel cell operating temperatures of 850 °C. However, 8YSZ can be difficult to sinter to 94% theoretical density, the density required for removing open porosity. Temperatures of 1400 °C or more for more than two hours are required to sinter 8YSZ to a density that makes it useful as a SOFC electrolyte [5]. Because of the usefulness and difficulty of processing of 8YSZ, many studies have been undertaken to understand why 8YSZ sinters at such a high temperature and how to reduce the temperature at which densification occurs.

To better understand the phenomenon of sintering, and find ways to increase sintering performance, the effects of initial powder size and shape have been examined [8,9,10,11]. Although improvements in sintering performance have been realized through changing sintering pressure, a function of powder particle size and shape, the sintering performance achieved via the addition of impurities as sintering aids can eliminate nano-particle requirements [12,13]. Al_2_O_3_ as a sintering aid has been shown to increase sintering performance by decreasing the activation energy of densification of 8YSZ [14,15]. A small amount of Al_2_O_3_ added to 8YSZ has been shown to increase ionic conductivity according to electrical impedance spectroscopy (EIS) [16]. The presence of Al_2_O_3_ at the electrolyte-anode interface was found to form NiAl_2_O_4_ by means of chemical interactions with Al_2_O_3_ and NiO [17]. Although the study results of Knibbe et al., suggested the presence of NiAl_2_O_4_ did not negatively impact electrochemical cell performance at the operating temperature of 850 °C, a study also performed by Knibbe et al., found that Al_2_O_3_ at the YSZ-La_1-x_Sr_x_MnO_3_ (LSM) interface caused delamination of the LSM air electrode from the YSZ, causing failure of the system [18].

The addition of some metal oxides may or may not act as a sintering aid, but have been shown to change the atomic structure of cubic zirconia. Adding TiO_2_ to cubic zirconia has been found to introduce the tetragonal phase of YSZ [19,20]. This observed phase change has been shown to decrease the ionic conductivity of YSZ [21]. The addition of Bi_2_O_3_ has also been found to destabilize the cubic phase of YSZ [22]. Nickel has been shown to enhance YSZ sintering performance [23]. However, nickel metal has been shown to precipitate out of YSZ solution in a reducing environment, irreversibly decreasing oxygen ion conductivity in the process [24,25,26]. Other effective sintering aids such as cobalt and manganese show promise as effective sintering aids that preserve the oxygen ion conductivity of 8YSZ [24,27,28].

The prior literature studies share the common theme of examining the effects of a single additive as a sintering aid. Reported 8YSZ sintering experiments concerned with more than one impurity typically examine the inadvertent addition of alumina or silica, and the negative performance impacts those additions have on electrochemical performance [29,30,31]. An in-house study concerned with the sintering performance of Ni(NO_3_)_2_ infiltrated 8YSZ in an oxygen rich atmosphere revealed that the Ni(NO_3_)_2_ infiltrated samples densified greatly in the oxygen-rich atmosphere.

Elemental investigation of the samples that demonstrated greater densification revealed the presence of large amounts of molybdenum, an impurity serendipitously present in the tube furnace from prior experimental work. It is thought that the molybdenum contamination of the furnace volatilized as molybdenum oxide in the oxygen rich atmosphere, providing a low-level source of well-distributed Mo. The nickel in the Ni(NO_3_)_2_ infiltrated pellets appears to have facilitated absorption of the molybdenum from the atmosphere, and the combination of nickel and molybdenum in the sub-micron particulate 8YSZ samples yielded substantial densification near 1100 °C, a temperature well below the onset of densification temperature of pure 8YSZ. This unexpected discovery inspired the study of the synergetic effects of nickel and molybdenum as sintering aids for 8YSZ.

## 2. Materials and Methods

Four sample groups were prepared for sintering studies: pure 8YSZ (1), Ni(NO_3_)_2_-infiltrated 8YSZ (2), 8YSZ with MoO_2_ (3), and Ni(NO_3_)_2_-infiltrated 8YSZ with MoO_2_ (4).

### 2.1. Control Group: Sample Set 1

8YSZ (TZ-8YS, Tosoh Corp., Tokyo, Japan) powder was uniaxially pressed as received in a cylindrical, stainless steel die under a pressure of 250 MPa. Samples were held under that pressure for 10 to 30 s to reduce damage from the powder springing back due to the storage of elastic energy in the compressed powder particles. Pressed pellets were then sintered in a vertical-load-frame dilatometer in a flowing oxygen atmosphere (L75, Linseis Messgeräte, Selb, Germany) up to 1200 °C with heating rates of 5 and 10 °C per minute, held at 1200 °C for one hour, then allowed to cool to room temperature. One of the sintering samples was fractured, attached to an aluminum slide with carbon tape, and iridium sputter coated. The microstructure of the pellet was then examined via scanning electron microscopy (SEM) (JSM-6100A SEM, Jeol Ltd., Tokyo, Japan). The remaining pellets were pulverized in a synthetic sapphire mortar and pestle. X-ray diffraction (XRD) (SCINTAG X1 Diffraction System, Scintag, Inc., Sunnyvale, CA, USA) was used to examine the sample crystal structure. Scans were conducted from 25°–78° 2θ with 0.01° step size and 3 s dwell.

### 2.2. Ni(NO_3_)_2_ Infiltrated Group: Sample Set 2

Pellets pressed in the manner described in Section 2.1 were infiltrated with a polymeric nickel-nitrate (Ni(NO_3_)_2_·6H_2_O, Inframat Advanced Materials, Farmington, CT, USA) solution prior to sintering. The polymeric nickel nitrate solution was prepared by dissolving Ni(NO_3_)_2_·6H_2_O in water. Ethylene glycol was added to the Ni(NO_3_)_2_ solution. The water was evaporated by holding the mixture at 95 °C while stirring. The solution was thinned with 2-n-butoxyethanol. The addition of nickel in this manner helped to insure uniform distribution of the addition particularly in the region of particle-particle interfaces. Pellets were infiltrated with polymeric Ni(NO_3_)_2_ solution by placing the pellet on a densified Al_2_O_3_ plate, and placing a few drops of Ni(NO_3_)_2_ solution onto the pellet. The pellet was allowed to absorb this solution for 24 h before it was sintered in the dilatometer in an atmosphere of flowing oxygen. This process resulted in a NiO content of approximately 0.85 mol.% after sintering. The NiO content of the samples was determined by weighing each pellet before infiltration with Ni(NO_3_)_2_ solution, and after sintering. The increased mass of the sintered pellets was attributed to the presence of NiO. A Mathcad algorithm which computed the mole fraction of NiO in the infiltrated pellet was used to determine the mole fraction of NiO in each sample. Sintering profiles described in Section 2.1 were used to sinter the Ni(NO_3_)_2_ infiltrated pellets. Samples were then examined via SEM and XRD according to Section 2.1.

### 2.3. Mechanically Mixed MoO_2_ 8YSZ: Sample Set 3

Three 8YSZ powders with MoO_2_ were prepared by mechanical mixing of oxides by dry ball milling appropriate amounts of molybdenum oxide powder (Molybdenum (IV) oxide, Alfa Aesar, Ward Hill, MA, USA) with 8YSZ in high density polyethylene bottles with zirconia milling media. The resulting powders contained 1.0, 5.0 and 10.0 mol.% MoO_2_. Each powder was then sieved through a screen before being pressed into pellets according to Section 2.1. A pellet of each concentration of MoO_2_ was then sintered in a flowing oxygen atmosphere using the sintering profiles described in Section 2.1. Samples were then examined via SEM and XRD according to Section 2.1.

### 2.4. Mechanically Mixed MoO_2_ 8YSZ Infiltrated with Ni(NO_3_)_2_: Sample Set 4

8YSZ pellets with each concentration of MoO_2_ were pressed according to the process described in Section 2.1. These pellets were then infiltrated with the polymeric Ni(NO_3_)_2_ solution according to the method described in Section 2.2. The pellet infiltration process resulted in pellets with 1, 5 and 10 mol.% MoO_2_ with about 0.85, 0.80 and 0.75 mol.% NiO, respectively. These pellets were sintered in a flowing oxygen atmosphere, and then examined via SEM, and XRD as described in Section 2.1.

### 2.5. Electrical Impedance Spectroscopy

Electrical impedance spectroscopy (EIS) was used to measure the electrical response of 8YSZ with the additions of 5 mol.% MoO_2_ and 1 mol.% NiO (Nickel (II) oxide, Alfa Aesar, Ward Hill, MA, USA). Powders were mixed according to Section 2.1. Pellets were pressed to 250 MPa, and sintered in air with a heating rate of 10 °C /min to 1350 °C and held for one hour. Note that EIS samples were not sintered in flowing oxygen for safety reasons, and that sintering in air required nickel to be added as NiO instead of Ni(NO_3_)_2_. Sintering Ni(NO_3_)_2_ infiltrated 8YSZ with MoO_2_ consistently led to pellets bursting during sintering.

EIS samples were sintered to 92% theoretical density. Measurements were made in air using a Solartron 1260 (Solartron Analytical, Wokingham, UK). Electrodes and leads were applied with Pelco High Performance Silver Paste (Ted Pella, Inc, Redding, CA, USA). The instrument was setup with a three-terminal connection with feedthrough terminators to prevent microwave back-reflection. Frequencies were swept from 1 Hz to 10 MHz with 71 logarithmically spaced points per decade. Data were collected from 350 to 500 °C in 25 °C increments. Samples were held at each temperature for 45 min to allow the attainment of thermal equilibrium. Datasets for analysis were chosen once sweeps stopped deviating significantly from one another.

## 3. Results

The time-density curves determined from the dilatometry measurements of the sintering performance of 8YSZ (Sample Set 1), 8YSZ infiltrated with Ni(NO_3_)_2_ (Sample Set 2), 8YSZ with MoO_2_ (Sample Set 3), and 8YSZ with MoO_2_ infiltrated with Ni(NO_3_)_2_ (Sample Set 4) are shown as Figure 1. Density was calculated from length change data according to Equation (1):(1)$$\rho \left(t\right)=\frac{\rho_{0}}{\rho_{th}{\left(1+\frac{\Delta L\left(t\right)}{L_{0}}\right)}^{3}}$$

Density as a function of time is given as *ρ*(t), initial density by *ρ*_0_ theoretical density by *ρ_th_*, length change as a function of time by Δ*L(t)*, and original sample length by *L*_0_. Each plot of Figure 1 directly compares the sintering performance of each sample set. Note that in all cases, the sintering rate and sintered densities of Sample Set 4 were much greater than the sintering performance of any other sample set. The maximum sintering rates, green pellet densities, and sintered pellet densities are summarized in Table 1. Note that densities are reported relative to the theoretical density of 8YSZ of 5.9 g/cm^3^. Sintering rates are reported in units of change in relative density per minute (%/min).

Figure 1A,B compare the sintering performance of 8YSZ, Ni(NO_3_)_2_ infiltrated 8YSZ, 1 mol.% MoO_2_ 8YSZ, and 1 mol.% MoO_2_ Ni(NO_3_)_2_ infiltrated 8YSZ sintered up to and held for one hour at 1200 °C with heating rates of 5° and 10 °C per minute, respectively. Plots C, and D of Figure 1 compare the sintering performance of 8YSZ, Ni(NO_3_)_2_ infiltrated 8YSZ, 5 mol.% MoO_2_ 8YSZ, and 5 mol.% MoO_2_ Ni(NO_3_)_2_ infiltrated 8YSZ sintered up to and held for one hour at 1200 °C with heating rates of 5° and 10 °C per minute, respectively. Plots E and F of Figure 1 compare the sintering performance of 8YSZ, Ni(NO_3_)_2_ infiltrated 8YSZ, 10 mol.% MoO_2_ 8YSZ, and 10 mol.% MoO_2_ Ni(NO_3_)_2_ infiltrated 8YSZ sintered up to and held for one hour at 1200 °C with heating rates of 5° and 10 °C per minute, respectively. The sintering temperature profile has been shown on each plot.

The sintering temperature of 1200 °C was intentionally chosen because of the poor sintering performance of 8YSZ at that temperature. The sintering performance increase that is achieved with the addition of 5 and 10 mol.% MoO_2_ with Ni(NO_3_)_2_ is greater than the sum of the sintering performance increases achieved with the individual addition of MoO_2_ and Ni(NO_3_)_2_. Further, it is shown that for each sample in Sample Set 4, the pellet sintered with a heating rate of 10 °C per minute achieved a greater final density than the pellet sintered with the 5 °C per minute heating rate.

The abrupt decrease in density of the 10 mol.% MoO_2_ Ni(NO_3_)_2_ infiltrated 8YSZ sample sintered at 5 °C per minute, as shown in the densification curve, is not fully understood. However, closer examination of the Sample Set 4 samples sintered at 10 °C per minute reveals similar, though less pronounced behavior.

Both densification and grain growth during sintering are Arrhenius processes. As such both behaviors are largely governed by the Arrhenius rate law, meaning that, in simple terms, the rate at which a phenomenon like densification or grain growth occurs is given by the product of a driving force, and a resistance term. The driving force of a phenomenon like sintering is related to the surface energy of the powder making up the initial powder compact. Since surface energy is proportional to surface area, the greater the surface area (the smaller the particles) of a powder compact, the greater the driving force for sintering.

The resistance term of this ubiquitous law is given as exp(−Q/RT). The quantity represented by Q is referred to as activation energy. In the case of a diffusion phenomenon like densification or grain growth, Q represents the amount of energy required for a cation to move from one equilibrium position to another. The quantity given by the product of the ideal gas constant R and absolute temperature T is the thermal energy.

As the ratio of the activation energy per thermal energy approaches 0 (as thermal energy becomes large when compared to activation energy), exp(−Q/RT) approaches 1. When activation energy is large when compared to thermal energy, exp(−Q/RT) is a number that is very close to zero.

The dramatic change in sintering behavior, shown in Figure 1, suggests activation energy for densification is significantly reduced with the addition of both Ni and Mo as sintering aids. Because the powder precursors are the same for all studies it is reasonable to assume the driving force of sintering, which depends on initial surface area of the particles that make up the powder compact is about the same.

If the driving force of densification remains constant, it is reasonable to hypothesize that the activation energy of both densifiation and grain growth are changing with the addition of these two sintering aids. However, a reduction in apparent activation enery is a non-specific result of a number of things that could be happening. Defect chemistry is a place to start explaining why two sintering aids might enhance sintering performance. However, the presence of molybdenum requires careful consideration, since it can assume cation valence states between +1 and +6. Thus, any defect chemistry equation would require vetting with x-ray photoelectron spectroscopy, and was deemed to be outside the scope of this work.

Figure 2 shows results of EIS study of 8YSZ and 8YSZ with the addition of 5 mol.% MoO_2_ and 1 mol.% NiO. Included as an inset is the Arrhenius plot of ionic conductivity associated with the 5 mol.% MoO_2_, 1 mol.% NiO 8YSZ. Concentrations of MoO_2_ and NiO were chosen based on sintering performance. Preliminary EIS results suggest a slight conductivity decrease with the addition of NiO and MoO_2_ to 8YSZ at 400 °C. This is comparable to Zhang et al.’s EIS measurements of 8YSZ with the addition of nickel as a sintering aid and 3000 ppm SiO_2_ [33].

Bulk ionic conductivity of the 5 mol.% MoO_2_ NiO 8YSZ was found to be $2.84\times 10^{-5}\;S/\mathrm{cm}$ at 400 °C. For comparison, Zhang et al. reports $1.26\times 10^{-4}\;S/\mathrm{cm}$ for 8YSZ, and $9.61\times 10^{-5}\;S/\mathrm{cm}$ for 8YSZ with the additions of 1 at% Ni and 3000 ppm SiO_2_ [34]. Activation energy for ion conduction in the bulk crystallite of 8YSZ with NiO and MoO_2_ was found to be 1.17 eV, whereas bulk activation energy of 8YSZ with 1 at.% Ni and 3000 ppm SiO_2_ reported by Zhang et al. was 1.17 eV.

Indexed results of X-ray diffraction studies and subsequent Rietveld refinement of pure YSZ, YSZ with Ni(NO_3_)_2_, and YSZ with 1, 5 or 10 mol.% MoO_2_, or 1, 5, or 10 mol.% MoO_2_ and Ni(NO_3_)_2_ are shown as Figure 3 [35,36]. Results suggest the addition of Ni(NO_3_)_2_, MoO_2_, and Ni(NO_3_)_2_, and NiO could lead to substitutional defects in the YSZ lattice. Results of the XRD study have been summarized as Table 2. 

Figure 3A shows a comparison between 8YSZ and 8YSZ infiltrated with Ni(NO_3_)_2_. Although differences between these two patterns are difficult to discern, Rietveld refinement and subsequent lattice parameter calculation suggests the addition of Ni(NO_3_)_2_ to 8YSZ decreases 8YSZ lattice parameter from 5.144 Å to 5.134 Å. Figure 3B compares XRD patterns of 1% MoO_2_ 8YSZ and 1% MoO_2_ 8YSZ with Ni(NO_3_)_2_. Figure 3C and 3D compare XRD patterns of 5% MoO_2_ 8YSZ and 5% MoO_2_ 8YSZ with Ni(NO_3_)_2_, and 10% MoO_2_ 8YSZ and 10% MoO_2_ 8YSZ with Ni(NO_3_)_2_, respectively. In each case, the addition of MoO_2_ to 8YSZ results in a decrease in 8YSZ lattice parameter while the addition of Ni(NO_3_)_2_ to 1, 5 or 10% MoO_2_ causes a slight lattice parameter increase, shown in Table 2.

Calculation of lattice parameters was done assuming the cubic structure of all samples was maintained, and were carried out according to Cullity [37]. Lattice parameters of 8YSZ (5.144 Å) and Ni(NO3)2 8YSZ (5.134 Å) are within 0.1% of published values of White et al. [38]. The addition of molybdenum is seen to change lattice parameters by about as much as the addition of nickel nitrate. However, results suggest the addition of 10 mol.% molybdenum oxide reduces the lattice parameter of cubic zirconia to 5.115 Å. Ionic radii of Zr^4+^ and Mo^4+^ with coordination numbers of 6 are given as 0.72 Å, and 0.65 Å, respectively. Thus, a reduction in lattice parameters of YSZ could be explained by substitution of Mo^4+^ ions on Zr^4+^ sites in the YSZ structure. Further experimentation, including XPS measurements, would be necessary to confirm this hypothesis.

The microstructure of Sample Sets 3 and 4 has been shown as Figure 4. Figure 4A shows the microstructure of 8YSZ with 10 mol% MoO_2_. This image suggests MoO_2_ may have little effect on both densification and grain-growth activation energy of 8YSZ. Figure 4B shows microstructure of 8YSZ with 1 mol% MoO_2_ and Ni(NO_3_)_2_. Grain growth is seen to be enhanced by the addition of both MoO_2_ and Ni(NO_3_)_2_, despite much less MoO_2_ in the sample shown as Figure 4B. The tiny rods shown in Figure 4C formed spontaneously after the 5 mol.% MoO_2_ Ni(NO_3_)_2_ infiltrated sample was left under vacuum overnight in the chamber of the SEM. It is reasonable to expect that such rods would not spontaneously form if such a material were used as an electrolyte in a solid oxide fuel cell, as fuel cells operate under conditions of flowing gases. An energy dispersive spectroscopy (EDS) line scan of one of the rods is shown as Figure 5D.

The microstructure of the 10 mol.% MoO_2_ Ni(NO_3_)_2_ infiltrated 8YSZ is shown as Figure 4D. Although the scales of Figure 4A,D allow for direct comparison of the two microstructures, comparison of Figure 4A with Figure 5A also demonstrates the effect on sintering behavior of using both MoO_2_ and Ni(NO_3_)_2_. All images of Figure 4 were taken in secondary electron imaging (SEI) mode.

Two possible molybdenum oxides, MoO_2_ and MoO_3_, are likely to form in an oxygen rich environment. Although the melting points of these molybdenum oxides are both below the sintering temperature of 1200 °C used in this work, the sintering performance of MoO_2_ 8YSZ was not better than the sintering performance of pure 8YSZ, which suggests volatilization of molybdenum oxide before the melting temperature was reached. Further, the porosity of the MoO_2_ 8YSZ samples, shown in Figure 4, suggests no liquid-phase sintering. Dramatic densification enhancement was only achieved with the addition of molybdenum and nickel.

The formation of a nickel-molybdenum compound has been considered to explain the increased densification. The only mention in the literature of such a compound is of NiMoO_4_. Material safety data sheets (MSDS) for NiMoO_4_ indicate that the melting point of this compound is not determined. However, the formation of this compound requires a nickel ion for every molybdenum ion. There is less than 1 mol.% NiO in each Ni(NO_3_)_2_ infiltrated pellet. Thus, the formation of NiMoO_4_ is stoichiometrically limited by the amount of nickel in the pellet, and it is not reasonable to conclude that liquid-phase sintering is responsible for the observed changes in sintering performance. Nickel is known to enter the lattice of 8YSZ [39].

Figure 5 shows effects of the MoO_2_ and Ni(NO_3_)_2_ addition on the microstructure of 8YSZ. Figure 5A shows a backscattered electron image (BEI) of 10 mol% MoO_2_ Ni(NO_3_)_2_ 8YSZ. Comparison of Figure 5A with Figure 4A suggests that the addition of Ni(NO_3_)_2_ to 8YSZ with 10∙mol.% MoO_2_ results in a microstructure with grain size of 50 μm, instead of a microstructure with nearly zero grain growth, and final grain size of about 0.5 μm. Figure 5B shows an EDS map of the same region as Figure 5A, and indicates grain boundaries are rich with molybdenum. Such grain growth was only observed with the 10 mol.% MoO_2_, Ni(NO_3_)_2_ infiltrated 8YSZ. Figure 5C shows the reference image of the line scan shown as Figure 5D. Said line scan indicates the spontaneously formed rods are made up of molybdenum and nickel. 

Images of the 1 and 5 mol.% MoO_2_ 8YSZ microstructures were not shown because of the similarity of the 1, 5, and 10 mol.% MoO_2_ 8YSZ microstructure. Contrasting the microstructure of Sample Set 3 with Sample Set 4 indicates that, by itself, the addition of MoO_2_ to 8YSZ does little to affect sintered microstructure of 8YSZ. This indication is confirmed by the similarity of the 1, 5, and 10 mol.% MoO_2_ images. Microstructures of the Ni(NO_3_)_2_ infiltrated 8YSZ, not shown, is similar to that of the 1, 5, and 10∙mol.% MoO_2_ 8YSZ microstructure.

Figure 4C and Figure 5C show the presence of tiny rods on the surface of the 5, and 10 mol.% MoO_2_, Ni(NO_3_)_2_ infiltrated samples, respectively. Elemental analysis of these rods indicates they are composed largely of nickel and molybdenum. The presence of these rods was not observed during the initial SEM analysis of the Ni(NO_3_)_2_ infiltrated samples with MoO_2_. The samples were left under high vacuum overnight to continue characterization the following day. These tiny structures were observed as SEM study continued. This observation suggests that the formation of these rods was entirely spontaneous.

## 4. Conclusions

The densification behavior, microstructure, and atomic structure of 8YSZ with MoO_2_ and Ni(NO_3_)_2_ additions were examined with dilatometry, XRD, and SEM. SEI was used to study ionic conductivity. Although the sintering performance of 8YSZ with either MoO_2_ or Ni(NO_3_)_2_ by itself was affected by the individual sintering aid addition, the effect on sintering of both of these materials together produced changes in the sintering performance that was seen to be greater than the sum of the performance increase of each individual sintering aid. Sintering rate increases of a factor of ten are observed in 8YSZ with additions of both MoO_2_ and Ni(NO_3_)_2_ over that of pure 8YSZ. Sintered densities of 85% theoretical density (5.9∙g⁄cm^3^) are achieved with the additions of MoO_2_ and Ni(NO_3_)_2_ at 1200 °C. This reflects a 57% increase in density over pure 8YSZ sintered with the same sintering profile.

Microstructural differences between 8YSZ with only the addition of MoO_2_ or Ni(NO_3_)_2_, and 8YSZ with both MoO_2_ and Ni(NO_3_)_2_ added suggest increased densification and grain growth with the inclusion of both sintering aids. Although a chemically induced phase change of 8YSZ may have been observed during the sintering of Ni(NO_3_)_2_ infiltrated 8YSZ with MoO_2_, the reasons for the enhanced sintering performance are not known.

EIS measurements suggest a low temperature decrease in ionic conductivity that is at least partially due to an increase in the activation energy of bulk ion conductivity. However, ionic Arrhenius results suggest ionic conductivity of 0.2 S/cm at 1000 °C. This value of ionic conductivity is slightly greater than the accepted value of 0.1 S/cm for pure 8YSZ at 1000 °C.

The observation that Ni and Mo work together as a sintering aid was entirely accidental. This serendipitous observation that molybdenum and nickel work well as sintering aids for 8YSZ in an oxygen rich environment suggest that other elements might work as synergistic sintering aids, perhaps without the need for sintering in flowing oxygen. Although the reason for this sintering performance enhancement is currently unknown (and is the subject of future work), understanding of this enhancement could lead to the deliberate development of other combinations of sintering aids for electrolyte materials that both enhance sintering performance and ionic conductivity.

## Figures and Tables

**Figure 1 materials-13-02875-f001:**
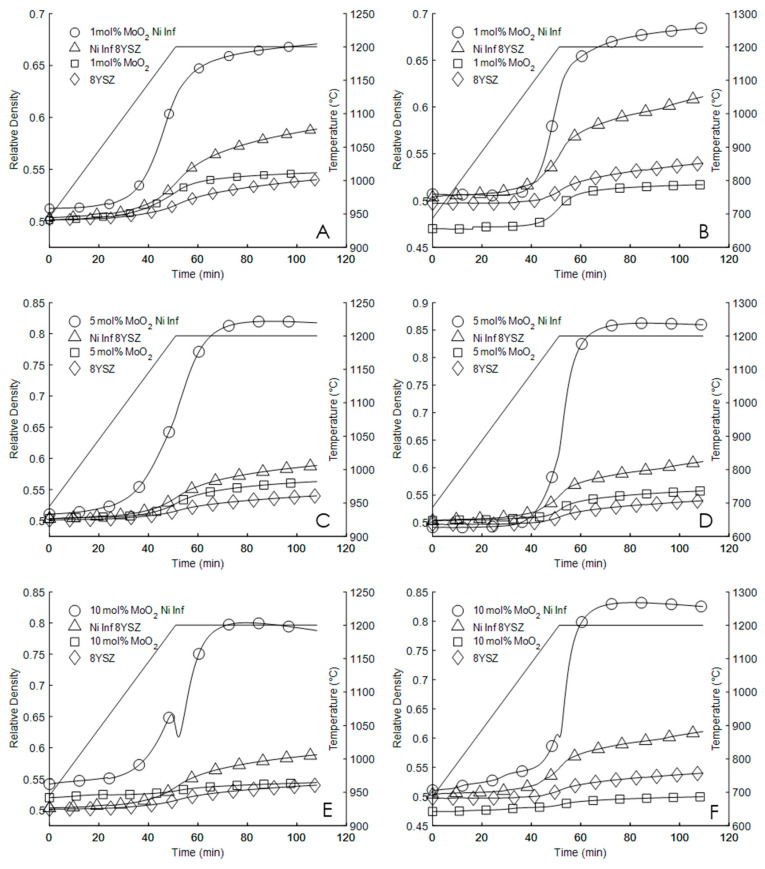
Sintering profiles of 8YSZ, Ni(NO_3_)_2_ infiltrated 8YSZ, 8YSZ with MoO_2_, and Ni(NO_3_)_2_ infiltrated 8YSZ with MoO_2_ sintered up to and held at 1200 °C for one hour with heating rates of 5 °C per minute (**A**,**C**,**E**), and 10 °C per minute (**B**,**D**,**F**) according to the temperature profile indicated in each figure by the solid black line (-). From [32], with copyright permission

**Figure 2 materials-13-02875-f002:**
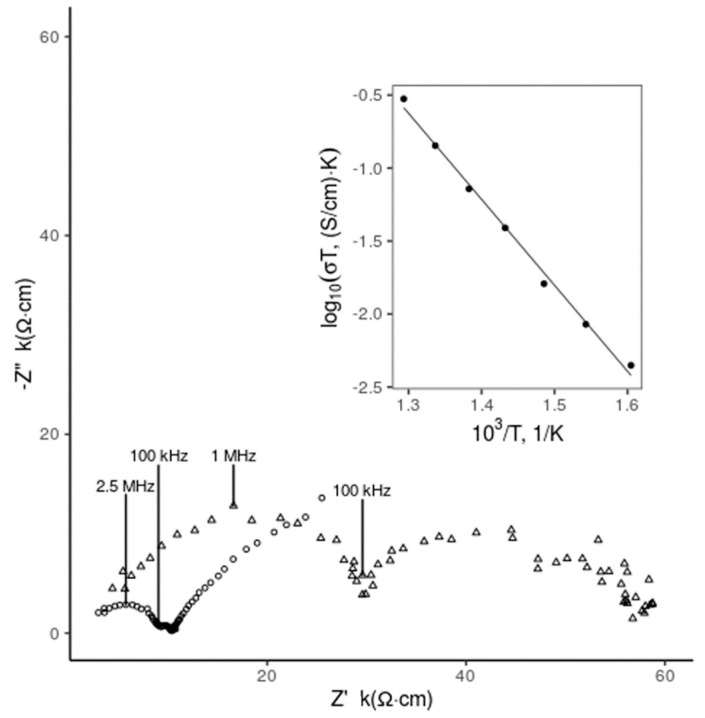
EIS spectrum of 8YSZ (○) and 8YSZ with the addition of 5 mol.% MoO_2_ and NiO as sintering aids (Δ) at 400 °C with Arrhenius conductivity plot as inset.

**Figure 3 materials-13-02875-f003:**
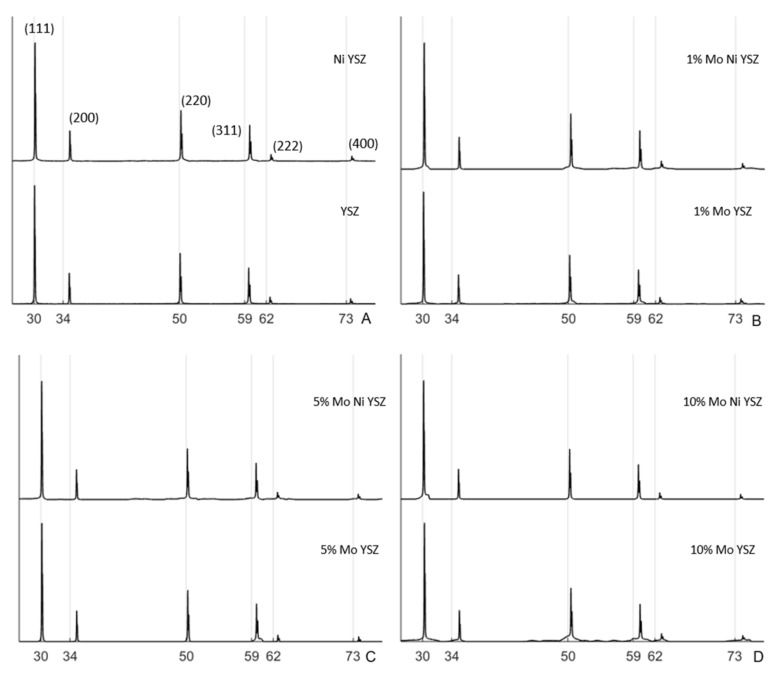
(**A**) XRD patterns with indexed peaks of cubic YSZ (YSZ), and cubic YSZ with Ni(NO_3_)_2_ infiltration sintered at 1200 °C for one hour, and XRD patterns of 8YSZ with the addition of (**B**) 1, (**C**) 5, and (**D**) 10 mol.% MoO_2_ and Ni(NO_3_)_2_ also sintered at 1200 °C for one hour. Each pattern is as indicated its plot. Peaks are indexed for cubic YSZ [35].

**Figure 4 materials-13-02875-f004:**
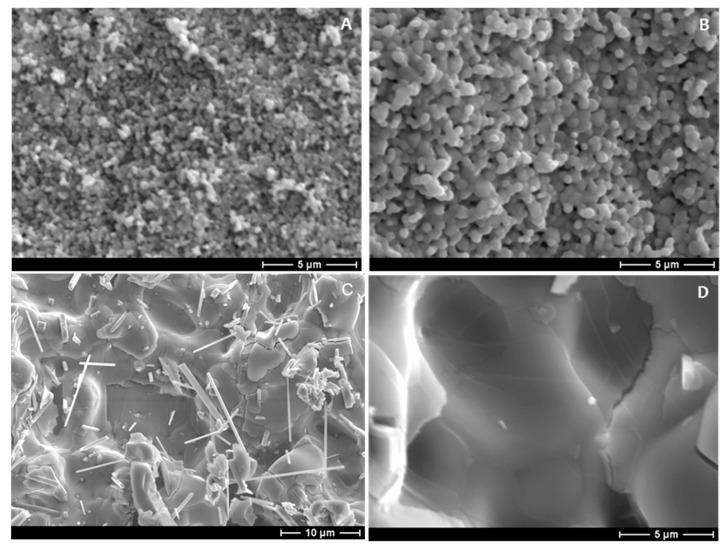
(**A**) 8YSZ with 10 mol.% MoO_2_. (**B**) Ni(NO_3_)_2_ infiltrated 8YSZ with 1 mol.% MoO_2_. (**C**) Ni(NO_3_)_2_ infiltrated 8YSZ with 5 mol.% MoO_2_. (**D**) Ni(NO_3_)_2_ infiltrated 8YSZ with 10 mol.% MoO_2_, where all samples were sintered up to and held at 1200 °C for one hour. From [32] with copyright permission.

**Figure 5 materials-13-02875-f005:**
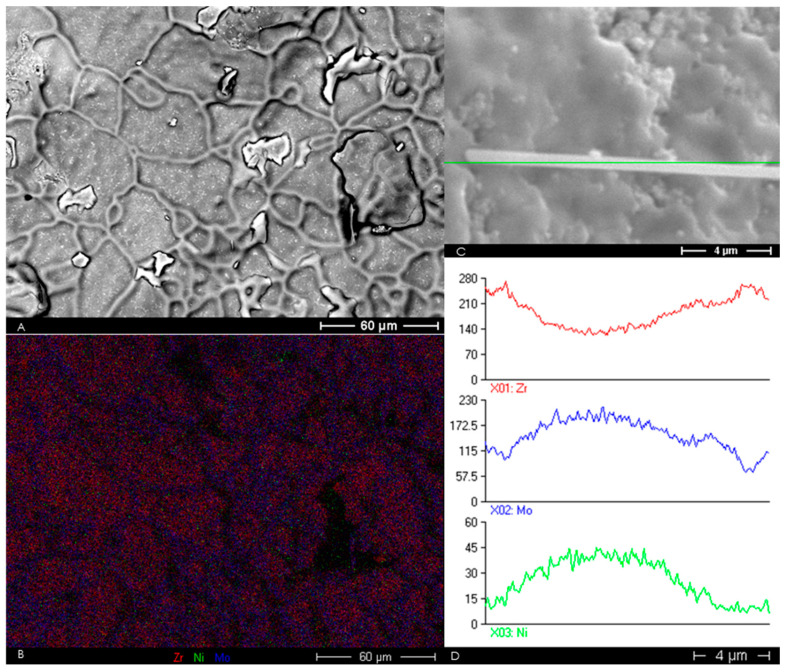
(**A**) 10 mol.% MoO_2_ Ni(NO_3_)_2_ infiltrated 8YSZ (BEI). (**B**) EDS map of region shown in A, suggesting Mo aggregates at grain boundaries. (**C**) Reference image of spontaneously formed nano-rod and (**D**) corresponding line scan indicating nano-rods are rich in Mo and Ni. From [32] with permission.

**Table 1 materials-13-02875-t001:** Green density, final sintered density, and maximum sintering rate; density measurements are relative to 5.9 g/cm^3^ and given as a percent of theoretical density, and sintering rate as change in percent relative density per minute (%/min) (Adapted from [32]).

Heating Rate (°C per Min)		Control	Ni(No_3_)_2_	1% MoO_2_	1% MoO_2_ Ni(No_3_)_2_	5% MoO_2_	5% MoO_2_ Ni(No_3_)_2_	10% MoO_2_	10% MoO_2_ Ni(No_3_)_2_
5	Initial Density	50.7	50.3	49.9	51.3	50.4	50.4	52.1	52.1
	Final Density	54.3	59.3	54.9	67.3	56.5	82	54.5	77.7
	Max Sintering Rate	0.101	0.249	0.154	0.718	0.204	1.287	0.068	2.227
10	Initial Density	50	50.4	47.2	50.7	49.6	49.2	47.7	51.6
	Final Density	54	61.2	51.9	69	55.9	85.9	50	82.4
	Max Sintering Rate	0.156	0.293	0.277	1.208	0.317	3.696	0.103	4.003

**Table 2 materials-13-02875-t002:** XRD peak locations determined according to Rietveld refinement, and associated lattice parameters of resulting structure assuming cubic symmetry.

	Peak Location (°2θ)						Lattice (Å)
	111	200	220	311	222	400	
YSZ	30.0632	34.8520	50.1140	59.5534	62.4907	73.5891	5.144
Ni YSZ	30.1260	34.9254	50.2234	59.6873	62.6327	73.7642	5.134
1%Mo YSZ	30.1106	34.9074	50.1965	59.6544	62.5978	73.7212	5.136
1%Mo Ni YSZ	30.1937	35.0046	50.3413	59.8317	62.7858	73.9530	5.136
5%Mo YSZ	30.1236	34.9226	50.2192	59.6822	62.6272	73.7575	5.134
5%Mo Ni YSZ	30.0991	34.8939	50.1764	59.6298	62.5717	73.6890	5.138
10%Mo YSZ	30.2407	35.0595	50.4232	59.9319	62.8921	74.0842	5.115
10%Mo Ni YSZ	30.1347	34.9356	50.2386	59.7058	62.6523	73.7884	5.132

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
