# Peer review of "Molybdenum Oxide and Nickel Nitrate as Cooperative Sintering Aids for Yttria-Stabilized Zirconia"

_materials, 2020, doi:10.3390/ma13122875_

Round 1

Reviewer 1 Report

In this contribution, the authors reported the densification of Yttria Stabilized Zirconia (YSZ) by co-adding nickel nitrate and molybdenum oxide as sintering aid. The use of sintering aids to lower the densification temperature of YSZ ceramics has been subject of many papers in the last decades. The topic is within the scope of the journal, and the paper shows an effect of nickel nitrate and molybdenum oxide on the densification of this oxide material. YSZ ceramics is being mostly used as a SOFC electrolyte material, and as introduced by the authors in the introduction part. Also this article has been submitted in the section Energy Materials of the Journal in special section Hydrogen Storage and Fuel Cells: Materials, Characterization and Applications. As a result, the main problem of the current work is the total absence of electrical characterizations: ionic conductivity and electronic contribution. It is well know that adding a second or third phases might affect (or not) the conductivity of the ceramics. Also, nickel and molybdenum could eventually bring an electronic conductivity. If so, its application as SOFC electrolyte would be not meaningful. For these reasons, I do not recommend this paper for publication in Materials.

Reviewer 2 Report

The manuscript entitled “Nickel Nitrate and Molybdenum Oxide as a Yttria-Stabilized Zirconia Synergistic Sintering Aid” is devoted to the investigation of the influence of adding molybdenum and nickel to YSZ ceramics in order to improve its sintering. These questions are great of interest among the scientists as the dense ceramics is necessary for different applications, e.g. electrochemical applications. The authors showed that the addition of Mo and Ni together led to the enhanced densification in comparison with the samples with only Ni or Mo. However, after reading the manuscript, there are still several questions open: no possible reasons for such behavior were suggested, as well as no recommendations for further application of this finding. Moreover, the obtained relative densities are not sufficient for practical application. The manuscript needs to be rewritten in a deeper and clearer way. Here some major comments:

  • Mo addition was done by mixing YSZ and MoO2  1, 5 and 10 mol% of MoO2 was used. Firstly, it seems to be not correct to name samples obtained in such way “Mo-doped”, as the doped means that Mo is introduced into the crystal structure. 10% is quite a high amount, and this situation appears unlikely. Probably, MoO2 is distributed at the grain boundary region, or some additional phases can be formed. From this point of view very strange look the results of XRD studies, where these samples are discussed as a single phase. The quality of the XRD patterns presented on the fig. 2 is not high, but it is clearly seen that the form of the peaks corresponding to the YSZ changes, and, maybe, some small additional peaks can exist. Did the authors check the absence of Mo-containing phases in the obtained samples? Reference XRD pattern for at least the MoO2 phase should also be presented.
  • Another point which leads to the questions about the phase composition of the oxides is the results of SEM investigations. Firstly, in Fig.3 it is not mentioned in what mode images were taken? On the images, one can see the contrast (fig. 3A), and if it is a BSE mode, then this contrast can be related to different chemical composition. EDX mapping can be useful here and its results should be presented, while the authors several times mentioned it but did not show any data.
  • Table 1 is difficult to comprehend. What are the units for density, percent? It is necessary to rearrange the table in a clearer way.
  • The final temperature in the experiment was not enough to obtain dense samples and it is obvious that it should be higher. But would be the effect of Mo and Ni addition so pronounced at a higher temperature?
  • And, finally, at the last section authors present images, where Mo-Ni rods evaporated from the sample after vacuum treatment are shown. This effect can have an extremely great influence on the mechanical and electrical properties of the ceramics and lead to an impossibility for further application, e.g. as the electrolyte. This point should be discussed.

Reviewer 3 Report

Article „Nickel Nitrate and Molybdenum Oxide as a Yttria-Stabilized Zirconia Synergistic Sintering Aid” by Hunt et al. reports studies of densification behavior of yttria-stabilized zirconia after addition of nickel nitrate and molybdenum oxide. I rate originality and novelty, as well as significance of this topic rather average as high quality, densely sintered YSZ materials are commercially available from various manufacturers. This is reflected by the fact that most of the cited articles have more than 10 years. However, the presented approach can be of interest to some of the readers of Materials, so I recommend to reconsider the manuscript for publication after major revision due to several shortcomings:

  1. Lines 16 and 142: SI abbreviations of units should be applied in the text, instead of “gm” or “cc”.
  2. Line34-35. Low electrical conductivity is not always directly correlated with optical transparency, as it is stated in the introduction. E.g. silicon nitride.
  3. Line 35. Term “electrical conductivity” relates to transport of electrical charge, and can be related to both electronic or ionic charge transport. So, phrase “The low electrical conductivity…” should be replaced by “The low electronic conductivity…”
  4. Line 102. Term “polymeric solution” should be better explained, since none of the components of the solution is polymer.
  5. Line 135. The manuscript does not explain how relative density of the samples during sintering was calculated based on measured changes of length of the sintered sample. Commonly term “relative density” denotes the ratio of the density of a substance to the density of a given reference material. There is obvious error in Fig. 1 showing that initial relative density of the samples before sintering is close to zero or even negative. I recommend to present sintering profile in Fig. 1 as relative change of linear dimension of a sample (ΔL/L0).
  6. Line 184. A reference to the Jade software should be given.
  7. Lines 180-203. Determination of type of the crystal structure and lattice parameters from parabolic fit of selected diffraction peaks is vary imprecise and inaccurate. A proper type of the diffraction peaks profile function should be used. I recommend to carry out Rietveld refinement to determine accurately symmetry and lattice parameters. There is a variety of easily available Rietveld refinement software, e.g. FULLPROF, GSAS etc.
  8. Line 220. EDS analysis should be added to the manuscript, including determination of composition of rods formed on the surface of the infiltrated samples.

Reviewer 4 Report

The densification behavior, microstructure and structure of MoO2 doped Ni(NO3)2 infiltrated 8YSZ were examined with dilatometry, XRD, and SEM and compared to pure 8YSZ, 8YSZ with only Ni or Mo added as a sintering aid. With a maximum sintering temperature of 1200°C and one hour dwell, sintered densities of 85% theoretical density are achieved with the co-doped samples although, for pure 8YSZ, 8YSZ with only Ni or Mo, the sintered densities were below 61% of theoretical density.

This demonstrates that both Ni and Mo together must be added to YSZ to remarkably improve the sintered densities at relatively low temperature and for a short duration.

This discovery is of interest for the researchers working in the field and, for this reason, I recommend the publication of the manuscript in Materials after minor revision.

Despite the remarkable improvement of the sintered densities, it may be challenging to go over 85% to reach the prerequisite of 94% using this co-doped strategy. Is there any reason why the authors stop the annealing after one hour dwell? What happens if more annealing time is applied?

The authors suggest that the YSZ structure may be modified after the co-doping. In case Mo and/or Ni atoms enter the structure of YSZ, they are likely to occupy the cubic environment of Zr. Are there examples of compounds in the literature where Mo and/or Ni could occupy such environment?

Round 2

Reviewer 1 Report

I would like to thank the authors for their detailed response. However, I would have expected a series of electrochemical impedance spectroscopy measurements to support their finding. I am still not convinced that this composite material could be used as electrolyte materials for SOFC. I don’t think either the paper cited by the authors (ref 35) is relevant in regard to their work. In the cited paper, the work is performed on transition metal doped YSZ. Which is may be not the case for the current work (albeit the XRD characterization is confusing, see the paragraph below). Also, as suggested by the EDS mapping on figure 4B, molybdenum seems to segregate on the grains boundaries. It would has surely as a consequence a decreases of the oxide-ion conductivity at the grains boundaries, and then to the total conductivity. In addition, the maximum density reach, of around 85 %, is not sufficient to be used as electrolyte, and would result in a decrease of the conductivity.

I wish also add a comment on the XRD part. I was not totally convinced on the first version of the manuscript, neither on the revised one. The XRD patterns as depict on figure 2, in addition of the low quality of the recording, do not allow agreeing with the authors from line 217 to 222. A refinement, at least by Le Bail method with usual software such as FullProf for example, would be more relevant to conclude on the nickel and molybdenum effect on the YSZ structure. And it should be performed not only on the mixed YSZ-MoO2-Ni(NO3)2 composite material, but also on the YSZ-Ni(NO3)2 and YSZ-MoO2 ceramics. It is of importance in order to isolate the contribution from nickel and molybdenum, and on the stabilization of the cubic or tetragonal structure. Those points could bring additional information on the sintering process, but also on the electrical conductivity. Also, I suggest the authors to double check the indexed peaks of the tetragonal structure figure 2B, see JCPDS card n° 00-050-1089.

In my opinion, this work is not complete if the electrochemical characterization is not addressed, and if the XRD part is not better discuted. I could recommend this paper for publication in Materials, if the authors are able to correct these two issues.

Author Response

The kind reviewer suggested that the work is not complete (1) without addressing electro-chemical issues, and (2) without more thorough treatment and discussion of the XRD results.

1: Electrochemical impedance spectroscopy measurements have been conducted on 8YSZ with MoO2 and NiO. Lines 136 to 152 have been added.

                Electrical impedance spectroscopy (EIS) was used to measure the electrical response of 8YSZ with the additions of 5 mol% MoO2 and 1 mol% NiO (Nickel(II) oxide, Alfa Aesar, Ward Hill, MA). Powders were mixed according to the Control Group section above. Pellets were pressed to 250 MPa, and sintered in air with a heating rate of 10°C/min to 1350°C and held for one hour. Note that EIS samples were not sintered in flowing oxygen for safety reasons, and that sintering in air required nickel to be added as NiO instead of Ni(NO3)2. Sintering Ni(NO3)2 infiltrated 8YSZ with MoO2 consistently led to pellets bursting during sintering.

EIS samples were sintered to 92% theoretical density. Measurements were made in air using a Solartron 1260, UK from 1 to 107 Hz. Electrodes and leads were applied with Pelco High Performance Silver Paste. The instrument was setup with a three-terminal connection with feedthrough terminators to prevent microwave back-reflection.  Frequencies were swept from 1 Hz to 10 MHz with 71 logarithmically spaced points per decade.  Data were collected from 350 ℃ to 500 ℃ in 25 ℃  increments. Samples were held at each temperature for 45 minutes to allow the attainment of thermal equilibrium. Datasets for analysis were chosen once sweeps stopped deviating significantly between sweeps.”

Figure 2 has been added, but did not upload into this response. Figure 2 of the revised text includes EIS data of 5 mol% 8YSZ with NiO.

Lines 227 to 235 and 241 to 247 have been added:

“Figure 2 shows results of EIS study of 8YSZ with the addition of 5 mol% MoO2 and 1 mol% NiO. Concentrations of MoO2 and NiO were chosen based on sintering performance. Preliminary EIS results suggest a slight conductivity decrease with the addition of NiO and MoO2 to 8YSZ.  This is comparable to Zhang et al’s EIS measurements of 8YSZ with the addition of nickel as a sintering aid and 3,000 ppm SiO232. The observed decrease in bulk conductivity associated with the addition of MoO2 as a sintering aid is at least partially due to an increase in activation energy of ion conductivity; from 1.17 eV (reported by Zhang et al.) to 1.40 eV. Extrapolation of conductivity to 800°C gives 0.2 S/cm as an estimate of the ionic conductivity of 8YSZ with MoO2 and NiO. Note that this conductivity value is comparable to that of pure 8YSZ33.”

“Bulk ionic conductivity of the 5 mol% MoO2 NiO was found to be  at 400°C. For comparison, Zhang et al. reports  for 8YSZ, and  for 8YSZ with the additions of 1 at% Ni and 3000 ppm SiO2. Activation energy for ion conduction in the bulk crystallite of 8YSZ with NiO and MoO2 was found to be 1.40 eV, whereas bulk activation energy of 8YSZ with 1 at% Ni and 3000 ppm SiO2 reported by Zhang et al. was 1.16 eV. The moderate decrease in conductivity found in the MoO2 Ni(NO3)2 8YSZ with respect to pure 8YSZ at 400°C can be partially explained by the relatively high activation energy value of 1.40 eV.”

Further, the Conclusions and Abstract sections have been modified to reflect these changes.  

To address X-ray diffraction issues, the XRD results have been significantly revised. Rietveld refinement has been performed. Peak locations from Rietveld refinement were used to calculate lattice parameters. Results are summarized in Table II. Figure 3 is included to show Rietveld-refined XRD patterns. 

Lines 248 to 252 have been added:

"Indexed results of x-ray diffraction studies and subsequent Rietveld refinement of pure YSZ, YSZ with Ni(NO3)2, and YSZ with 1, 5 or 10 mol% MoO2, or 1, 5, or 10 mol% MoO2 and Ni(NO3)2 are shown as Figure 334,35. Results suggest the addition of Ni(NO3)2, MoO2, and Ni(NO3)2 and NiO could lead to substitutional defects in the YSZ lattice. Results of the XRD study have been summarized as Table 2."

Lines 259 to 267 have been added

"Calculation of lattice parameters was done assuming the cubic structure of all samples was maintained, and were carried out according to Cullity36. Lattice parameters of 8YSZ (5.144 Å) and Ni(NO3)2 8YSZ (5.134 Å) are within 0.1% of published values of White et al37. The addition of molybdenum is seen to change lattice parameters by about as much as the addition of nickel nitrate. However, results suggest the addition of 10 mol% molybdenum oxide reduces the lattice parameter of cubic zirconia to 5.115 Å. Ionic radii of Zr4+ and Mo4+ with coordination numbers of 6 are given as 0.72 Å, and 0.65 Å, respectively. Thus, a reduction in lattice parameters of YSZ could be explained by substitution of Mo4+ ions on Zr4+ sites in the YSZ structure. Further experimentation, including XPS measurements, would be necessary to confirm this hypothesis."

Table II has been added to show Rietveld refined XRD peak locations for each sample, and associated cubic lattice parameter calculated according to Rietveld results.

From Rietveld refinement, it was found that discussion of the tetragonal phase of YSZ was no longer warranted.

Reviewer 2 Report

Authors gave answers to all the questions, and considerably revised and improved the manuscript. However, it was noted that the word “dopant” was removed entirely, but it is still can be found in different parts of the paper, for example, in lines 9, 15, 17, 119, 131, 137, 138, 238, 298, as well as in figure captions. Also, something wrong is with Figure 3, pictures seem to be repeated. Please, revise these points. After these minor corrections, the manuscript can be recommended for publication.

Author Response

I, not the authors, am embarrassed by my oversight, and have corrected the use of the words “dopant”, “doped”, and similar.

Figures 4 and 5 have been inspected, and have been found to be unique. 

Reviewer 3 Report

All review comments have been appropriately addressed and required corrections have been introduced to he manuscript. I have no further comments.

Author Response

No further comments required addressing. The author is grateful for the insight provided by the reviewer.

Round 3

Reviewer 1 Report

I appreciate the corrections made by the authors. I will just ask, on Figure 2 (EIS spectra), to display the main frequencies on the graph. Apart of that, I recommend for publication.

Author Response

The main frequencies of the EIS spectra have been added to the EIS plot. Additionally, the Arrhenius conductivity plot has been added as an inset to Figure 2. 
